# Hydrodynamic and Mass Transfer in the Desorption Process of CO$_2$ Gas in a Packed-Bed Stripper

**Pao Chi Chen [1,*], Ming-Wei Yang [2] and Yan-Lin Lai [1]**

[1] Department of Chemical and Materials Engineering, Lunghwa University of Science and Technology 300, Sec. 1, Wanshou Rd., Guishan District, Taoyuan City 33306, Taiwan; xx94u04xup6g4gj94ek@gmail.com
[2] Research Institute of Taiwan Power Company 84, Da-an Rd., Shu-Lin Dist., New Taipei City 23847, Taiwan; u620967@taipower.com.tw
[*] Correspondence: chenpc@mail2000.com.tw

**Abstract:** A lab-scale packed-bed stripper containing Dixon rings was used to explore the effects of the process variables on the hydrodynamics and mass-transfer in a stripper using a mixed solvent loaded CO$_2$. The variables are the liquid flow rate, reboiler temperature, and amine concentration, and the hydrodynamic and mass-transfer data can be determined using different models. In the case of hydrodynamics, the dimensionless pressure drop at the flooding point and the total pressure drop were explored first. In the case of mass-transfer, the correlation of the mass-transfer coefficient and the parameter importance were also observed. In addition, the number of plates per meter can be compared with the Dixon rings manufacturer. Finally, the performances of a mixed solvent and monoethanolamine (MEA) solvent were also discussed.

**Keywords:** stripper; hydrodynamic; mass transfer

## 1. Introduction

Descriptions of the hydrodynamics of a packed bed are generally dominated by a channel model and particle model [1]. The former assumes that air flows upward through several similar channels, while liquid flows downward along the channels against the channel wall. As liquid flows against the channel wall, the section for air flowing upward is reduced, thus, the pressure drop is increased. The particle model assumes that air flows around the particles, during which, the effective dimensions of particles are increased as liquid is attached to the particles, and the void ratio is reduced. Similarly, the particle model can be applied to the CO$_2$ stripping system, as shown in Figure 1a. As there is particle absorption heat energy from the steam and released CO$_2$ within the particle, steam and CO$_2$ gases flow up simultaneously, thus, this model can be described as a two-film model, as shown in Figure 1b.

Stichlmair et al. [1] applied the particle model and the gas flow rate and void ratio model in a fluidized bed developed by Richardson and Zaki [2], as well as a pressure drop computation equation for forecasting the flow phenomena of a packed bed, including the loading point, loading region, and flooding, and the forecasted values fit the experimental values well. Thereafter, Rocha et al. [3] developed a flow pattern model of a distillation column by applying structured packings for forecasting the liquid holdup within the bed, the pressure drop, and the flooding capacity, and such a model has also been validated through air/water and organic components distillation systems, with operating pressure ranging from 0.02 to 4.14 bar. Afterwards, Hoffmann et al. [4] published a paper that proposed the scaling-up of a reaction distillation column with a catalyst. For the purpose of this physical system, knowledge of reaction kinetics, phase equilibrium, and packing characteristics is required for determining the pressure drop, liquid holdup, and separation efficiency. The rate-based model was introduced in the proposed new hydrodynamic model to describe the gas–liquid contact flow pattern within the full load range, which also

considered the impact of the specific surface area, void ratio, and catalytic agent volume fraction on the bed diameter.

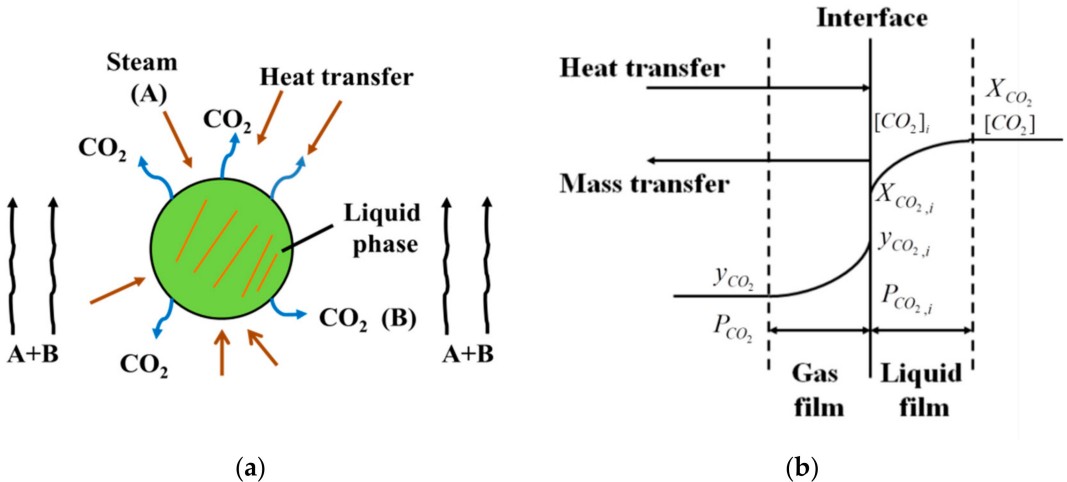

**Figure 1.** Steam and liquid drop contact within the packed-bed stripper. (**a**) Particle model, (**b**) two-film model.

Subsequently, Rocha et al. [5] developed a mass transfer model based on the previously developed flow pattern model. This model covered the effective surface area and provided an empirical equation for forecasting mass transfer efficiency under different types of packings, different flow conditions, and the different physical properties of fluids, as represented by the height equivalent to a theoretical plate (HETP), and the theoretical values fit the forecasting values well. In the following year, Guolito et al. [6] developed a design method for a distillation column of structured packings, which was mainly attributed to the higher efficiency, lower pressure drop, and greater capacity of the distillation column of the structured packings, and discussed changes in HETP with the pressure under different pressure levels. They also considered further revising the model, in order to obtain more accurate forecasting values. In view of the above, while there is still a difference between the packed bed model applying structured packings and the actual conditions, relevant scholars are making efforts to make advancements to improve the application value of the model; for example, Ortiz-Del Castillo et al. [7] proposed a desorption column applying random packing and structured packing to design a desorption column of organic components.

In this study, a packed-bed stripper with mixed solvent (monoethanolamine (MEA) + 2-amino-2-methyl-1-propanol (AMP)) loaded $CO_2$ was used to desorb $CO_2$ to explore the effect of process parameters, such as the concentration of blended amine ($C_A$), feed rate ($Q_L$), and the temperature of the reboiler ($T_{reb}$) on the hydrodynamics, such as pressure drop, flooding point, and F-factor, as well as mass transfer, such as the mass-transfer coefficient and height of transfer unit (HTU), and a base-line of MEA solvent was also adopted for comparison. In order to determine hydrodynamic data and mass transfer data, the models proposed in literature [7], including a hydrodynamic model and mass transfer model, were applied, mainly because such models applied desorption of volatile organic solvents, while this study applied $CO_2$ for desorption. Both models applied vapor desorption with similarities. In order to explore hydrodynamic and mass transfer in the desorption of $CO_2$, the thermodynamic data, such as viscosity [8,9], diffusivity [9,10], density [8,11], surface tension [12,13], Henry's law constant [14,15], and heat capacity [16], had to be determined. Figure 2 illustrates the framework of the research project, where the steps involve the process variables, materials, and energy balances, thermodynamic calculation, hydrodynamic calculation, mass transfer calculation, and verification of the models.

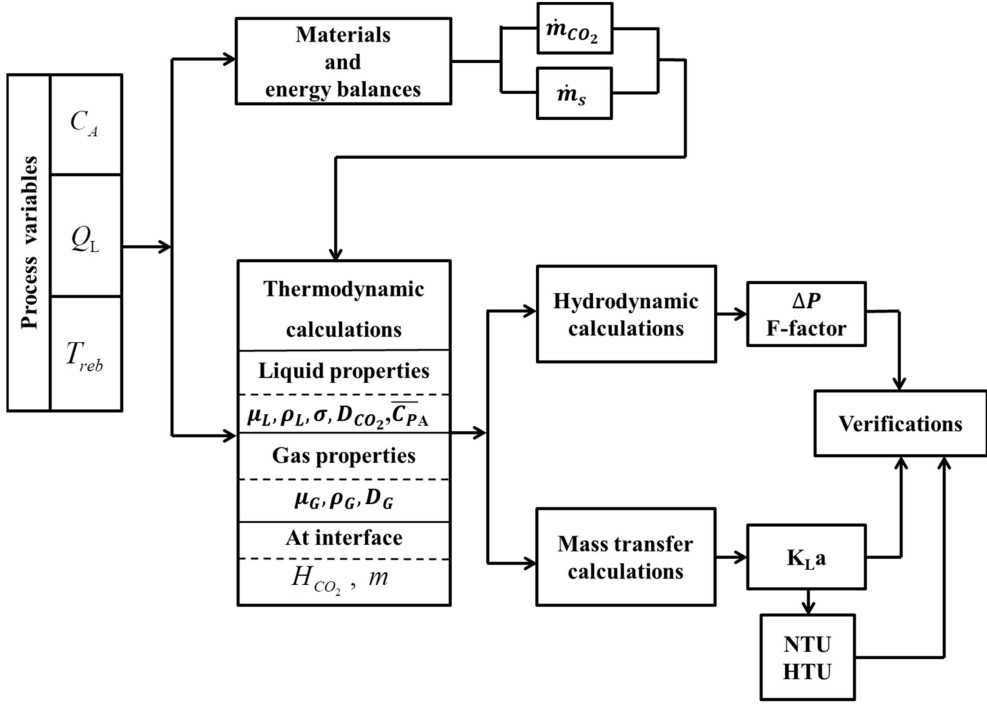

**Figure 2.** The framework proposed in this research.

## 2. Models

In the stripping process, rich solvents are heated in the stripper to allow the release of $CO_2$ gas from the scrubbed solutions. The stripping vapor, which involves water vapor, $CO_2$ gas, and small amounts of solvents, is regenerated in the reboiler, and then rises from the reboiler through the column to the top of the stripper. The stripping vapor counter-current contacts the rich-loading feed stream, which absorbs energy from the stripping steam for $CO_2$ gas desorption, while the remaining vapor is condensed at the top of the column in an overhead condenser. The design of the stripper must consider the hydrodynamic and mass-transfer coefficient of the packed-bed column, as the hydrodynamic calculations provide the diameter of the column, while the mass-transfer calculations provide the effective height of the column. During the operation, the mass, heat, and momentum transport occur simultaneously; therefore, both phenomena should be considered together for modeling. During the gas–liquid contact, the liquid holdup ($h_{dyn}$) is the parameter of effective velocities for the linkage of mass-transfer and momentum balances. For random packings, the gas and liquid effective velocities can be evaluated, as follows:

$$u_{Ge} = \frac{u_G}{\varepsilon(1 - h_{dyn})} \tag{1}$$

$$u_{Le} = \frac{u_L}{\varepsilon h_{dyn}} \tag{2}$$

The model parameters used in the above are listed in Equations (3)–(15) [7].

$$d_p = \frac{6(1 - \varepsilon)}{a_p} \tag{3}$$

$$u_G = \frac{Q_G}{A} \tag{4}$$

$$Re_G = \frac{u_G \rho_G d_p}{\mu_G} \tag{5}$$

$$u_L = \frac{Q_L}{A} \tag{6}$$

$$\psi = \frac{0.05}{\text{Re}_G} + \frac{1}{\text{Re}_G^{1/2}} + 3 \tag{7}$$

$$\Delta P_{dry} = \frac{1}{8} \psi a_p \frac{\rho_G u_G^2}{\varepsilon^{0.45}} \tag{8}$$

$$h_{dyn} = 3.6 \left( \frac{u_L a_p^{0.5}}{g^{0.5}} \right)^{0.66} \left( \frac{\mu_L a_p^{1.5}}{\rho_L g^{0.5}} \right)^{0.25} \left( \frac{\sigma a_p^2}{\rho_L g} \right)^{0.1} \tag{9}$$

$$h_{dyn} = h_{dyn0} \left[ 1 + \left( 6 \frac{\Delta p_{tot}}{\rho_L g} \right)^2 \right] \tag{10}$$

$$d_L = 0.4 \sqrt{\frac{6\sigma}{\Delta \rho g}} \tag{11}$$

$$a_L = \frac{6 h_{dyn}}{d_L} \tag{12}$$

$$\frac{\Delta p_{tot}}{\Delta p_{dry}} = \frac{a_L + a_p}{a_p} \left( \frac{\varepsilon}{\varepsilon - h_{dyn}} \right)^{4.65} \tag{13}$$

$$\frac{\Delta P_{tot,flood}}{\rho_L g} = \frac{\sqrt{249 h_{dyn0}(\sqrt{X} - 60\varepsilon - 558 h_{dyn0} - 103 d_L a_p)}}{2988 h_{dyn0}} \tag{14}$$

$$X = 3600\varepsilon^2 + 186480 h_{dyn0}\varepsilon + 32280 d_L a_p \varepsilon + 191844 h_{dyn0}^2 + 95028 d_L a_p h_{dyn0} + 10609 d_L^2 a_p^2 \tag{15}$$

Using this model, the liquid holdup, pressure drop, and pressure drop at flooding can be easily estimated using an Excel computer program. In addition, the mass-transfer model was first proposed by Gualito et al. [6] and modified by Ortiz-Del Castillo et al. [7], as shown in Equations (16)–(23) [7]. The differences of these models in their individual mass-transfer coefficients are the effective velocity of the former and the relative effective velocity of the latter. In a comparison mass-transfer coefficient, only one expressed the Schmidt number; whereas Schmidt and Reynolds numbers were considered in Equations (16) and (17). In addition, the coefficient and exponent are different.

$$k_G = \frac{0.1 D_G}{d_p} \left[ \frac{d_p (u_{Ge} + u_{Le})\rho_G}{\mu_L} \right]^{0.2405} \left[ \frac{\mu_G}{\rho_G D_G} \right]^{1/3} \tag{16}$$

$$k_L = \frac{0.3415 D_L}{d_p} \left[ \frac{d_p (u_{Ge} + u_{Le})\rho_L}{\mu_L} \right]^{0.2337} \left[ \frac{\mu_L}{\rho_L D_L} \right]^{1/2} \tag{17}$$

$$\frac{a_e}{a_p} = \frac{\left[ \left( \frac{u_L^2 \rho_L d_P}{\sigma} \right) \left( \frac{u_L^2}{g d_p} \right) \right]^{0.15} a_p d_p^{\,C}}{\left[ \frac{u_L \rho_L d_p}{\mu_L} \right]^{0.2} \varepsilon^{0.6}(1 - \cos \gamma)} \tag{18}$$

$$\frac{1}{K_L a_e} = \frac{1}{k_L a_e} + \frac{1}{(k_G a_e)m} \tag{19}$$

$$HTU = \frac{u_L}{K_L a} \tag{20}$$

$$NTU = \frac{S}{S-1} \ln \left[ \left( \frac{S-1}{S} \right) \frac{x_{in}}{x_{out}} + \frac{1}{S} \right] \tag{21}$$

$$S = \frac{mG}{L} \tag{22}$$

$$m = \frac{H_{CO_2}(\frac{kpa \cdot m^3}{kmol})C(\frac{Kmol}{m^3}) \times 10^3 (pa/kpa)}{P(atm) \times 1.0325 \times 10^5 (pa/atm)} \tag{23}$$

## 3. Experiment

### 3.1. Experimental Design

The experimental design has three factors: The concentration of mixed amine (MEA + AMP), the feed rate, and the reboiler temperature. Originally, MEA and AMP (30 wt% AMP in total amine) were mixed together; then, the mixed amines were poured into a known amount of water to prepare the desired amine concentrations. The mixed amine concentrations were 3 $kmol/m^3$, 4 $kmol/m^3$, and 5 $kmol/m^3$; the feed rates were 0.2, 0.3, and 0.4 L/min; the reboiler temperatures were 100, 110, and 120 °C, and the $CO_2$ loading was 0.4 kmol-$CO_2$/kmol-amine, respectively. The MEA/AMP weight fraction ratios obtained here were 0.1421/0.0690, 0.1908/0.0818, and 0.2388/0.1023 for 3 $kmol/m^3$, 4 $kmol/m^3$, and 5 $kmol/m^3$, respectively. Table 1 shows the operating conditions in this work from No. 1 to No. 12, while No. 13–No. 15 use an MEA solution as a baseline for comparison. The rich loading for the feed solution was obtained by early experimental preparation.

**Table 1.** Operating conditions conducted in this work.

| No | T (°C) | $Q_L$ (L/min) | $C_A$ (kmol/m$^3$) |
|----|--------|---------------|---------------------|
| 1  | 100    | 0.2           | 3                   |
| 2  | 100    | 0.3           | 3                   |
| 3  | 100    | 0.4           | 3                   |
| 4  | 110    | 0.2           | 4                   |
| 5  | 110    | 0.3           | 4                   |
| 6  | 110    | 0.4           | 4                   |
| 7  | 120    | 0.2           | 5                   |
| 8  | 120    | 0.3           | 5                   |
| 9  | 120    | 0.4           | 5                   |
| 10 | 110    | 0.3           | 3                   |
| 11 | 110    | 0.3           | 4                   |
| 12 | 110    | 0.3           | 5                   |
| 13 | 110    | 0.3           | 3                   |
| 14 | 110    | 0.3           | 4                   |
| 15 | 110    | 0.3           | 5                   |

### 3.2. Experimental Device and Operating Procedure

The stripping system is illustrated in Figure 3, including a packed column, a reboiler, a condenser, a heat exchanger, and a heating system; the diameter of the column and length are 50 mm and 800 mm, respectively; the height of the condenser is 500 mm; the packed column is filled with 8 × 8 mm Dixon packings (θ-ring); a 12 L reboiler was used to generate steam heating by the silicone oil using heating tubes, and a pressure back valve was adopted as a suitable value at the top and bottom of the column. To start the system, the temperature indicators, cooling water circulator, and oil-bath power supply were switched on and adjusted to a preset temperature. Second, the prepared rich loading solution was poured into the reboiler until it flooded, and when the oil-bath temperature reached the set point, the oil-bath pump's power supply was turned on, and an oil-bath valve was used to adjust the oil flow. Third, the inlet temperature of the cooling water was set to the desired condition, the reboiler was regulated to the desired temperature, and the experiment started when these temperatures reached the set points. The rich loading in the storage tank flowed through a heat exchanger and into a packed bed, and came in contact with the vapor rising from the bottom of the column. The lean loading was withdrawn at the bottom of the reboiler and passed through the heat exchanger, which released the heat to the rich loading as the input solvent, and the lean loading was withdrawn every 30 min for sample examination using the titration method. In addition, the flow meter was

calibrated using a measuring cylinder; then, a micro adjustment was made to the desired value. During the experiment, all temperature points indicated in Figure 3 were recorded, including the bed temperature (T01–T05).

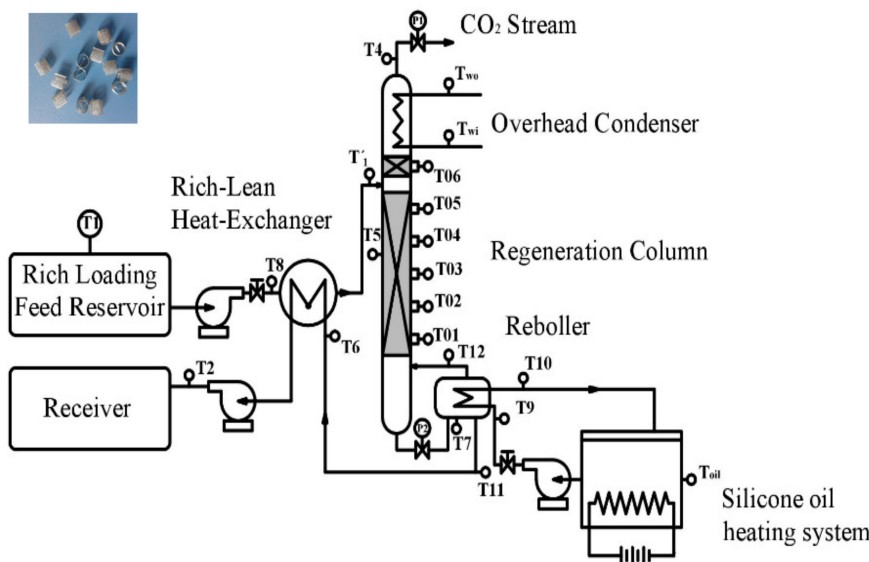

**Figure 3.** A stripping process with θ-ring packings conducted in this work [17]. T1: Inlet temperature of rich loading; T2: Temperature at outlet of heat exchanger; T4: Out temperature at the top of cooler; T5: Temperature of bed; T6: Liquid temperature at the inlet of heat exchanger; T7: Reboiler temperature; T8: Inlet temperature of rich loading; T9: Oil temperature at the inlet of reboiler($T_{in}$); T10: Oil temperature at the outlet of reboiler($T_{out}$); T11: Outlet temperature the reboiler; T12: Steam temperature; $T_{wi}$: Inlet temperature of cooling water; $T_{wo}$: Outlet temperature of cooling water; $T'1$: Inlet temperature at top of the column; P1: Pressure valve at the top of tower; P2: Pressure valve at the bottom of tower.

## 4. Results and Discussion

### 4.1. Dynamic and Steady State of the CO$_2$ Stripper

Variations in temperature for No. 1 were observed and recorded during operation, as shown in Figure S1. Several important points were recorded, such as $T1$, $T'1$, $T2$, $T7$, and $T12$, as presented in this figure. Here, $T1$ and $T2$ are the inlet and outlet temperatures of the liquids, respectively; $T'1$ is the input temperature at the top of the column; $T7$ and $T12$ are the reboiler temperature and steam temperature, respectively. It was found that the temperatures were kept nearly constant when the operating time was greater than 100 min. In addition, the temperature distribution in the packed bed was recorded, as shown in Figure S2. The distributions approached a steady-state operation after 100 min, and the distribution of the lean loading was found to remain constant after 100 min (see dotted line), as shown in Figure S3, which is coincident with the data in Figures S1 and S2. Thus, it could be said that the system changed from a dynamic state to a steady state after 100 min. All data could be calculated when a steady state was reached, and the data are shown in Tables S1–S4 and discussed later.

### 4.2. Hydrodynamic in a Packed Column

The flow velocity of a fluid, including the superficial velocity within the packed bed, must be noted if the flow pattern within the packed bed is to be understood. Therefore, relevant data on packing, including the void fraction ($\varepsilon$) and specific surface area ($a_p$), are required, and this was quite complicated due to gas–liquid contact within the bed. Furthermore, the relation between the flooding point of the fluid and the pressure drop was also data required for operating conditions and design. This study also collected research data on the hydrodynamics within the packed bed [1,4,7], including the effective diameter ($d_p$),

the Reynolds number ($Re_G$), pressure drop ($\Delta P$), liquid holdup ($h_{dyn}$), droplet diameter ($d_L$), liquid specific surface area ($a_L$), and flooding pressure drop ($\Delta P_{tot,flood}$). Calculations based on known data are shown in Equation (3) to Equation (15), and calculation data are shown in Table S2. The effective diameter is $3.273 \times 10^{-4}$ m; the gas flow rate, as shown in twelve sets of data, is 0.3572–1.1133 m/s; the liquid flow rate is 0.0017–0.0034 m/s; the dry-bed pressure drop is 23.3–276.2 Pa/m; the liquid dynamic holdup is 0.048–0.080, the holdup under the loading point is 0.050–0.083, the total pressure drop recorded is 78.8–528.9 Pa/m; and the pressure drop at the flooding point recorded is 1360–1725 Pa/m. It can be seen that the operating pressure drop in this study is less than the flooding pressure drop, indicating that there is no risk of flooding in the operation. The ratios of the individual total pressure drop to the flooding pressure drop range from 0.0487 to 0.346. All the data indicated in Table 2 are discussed later.

**Table 2.** A comparison of mixed solvent with base-line solvent in hydrodynamic and mass-transfer. HTU: Height of transfer unit.

| Experimental Number | $\dot{m}_{CO_2}$ (g/min) | $\dot{m}_s$ (g/min) | $\Delta P_{tot}$ (pa/m) | $\Delta P_{tot,flood}$ (pa/m) | $k_L a$ (1/s) | HTU (m) |
|---|---|---|---|---|---|---|
| No.13–15 (MEA) | 8.4–11.6 | 119.8–147.7 | 799.9–1150 | 1441–1492 | 0.01–0.0152 | 0.1678–0.2535 |
| No.10–12 (MEA + AMP) | 7.82–11.7 | 66.1–107.6 | 257.1–528.9 | 1514–1562 | 0.0244–0.0261 | 0.0976–0.1045 |
| No.1–12 (MEA + AMP) | 3.77–15.3 | 34.8–107.6 | 83.4–528.9 | 1360–1725 | 0.0187–0.0272 | 0.0823–0.1329 |

The plots of $\Delta P$ versus parameters under various conditions are shown in Figure 4a,b. It was found that the pressure drops at the flooding point in Figure 4a were close together and decreased with $Q_L$, while the $\Delta P_{tot}$ was different under different conditions and increased with the increase in $Q_L$. In addition, in order to compare the difference between the MEA and MEA + AMP solvents, the plot of $\Delta P$ versus $C_A$ is shown in Figure 4b. It was found that the pressure drops at the flooding point were close together for both, while the $\Delta P_{tot}$ for MEA was higher than that for MEA + AMP, which was due to the vapor flow rate ($\dot{m}_s$) being higher than the mixed amine, as shown in Table S1 (No. 10–No.15). It can be said that the mixed amine was more flexible in the pressure drop, as compared with the MEA solvent. In addition, it was found that the pressure drop for MEA was close to the pressure drop at the flooding point and increased with $C_A$, becoming flooded when the concentration was higher than 5 kmol/m$^3$.

However, as stated in Ref. [1], most of the pressure drops at the flooding point fall within, which sets as guidelines rules:

$$\frac{\Delta P_{tot,flood}}{\rho_L g} = 0.1 - 0.3 \tag{24}$$

while it has been noted from the data of this paper that the flooding point falls within 0.142–0.181, it is consistent with literature.

Further, there is a certain relation between $\Delta P_{tot}$ and the F-factor, which is defined below [18]:

$$F = u_{Ge}\sqrt{\rho_G} \tag{25}$$

and could be an indicator for flooding. Generally, its upper limit is 2.4 [19], and there may be a risk of flooding if it exceeds the value. Figure 5 shows $\Delta P_{tot}$ based on the F factor. A log-log graph is shown in Figure 5, and the following relationship can be noted:

$$\Delta P_{tot} = 397.2 F^{2.03} \tag{26}$$

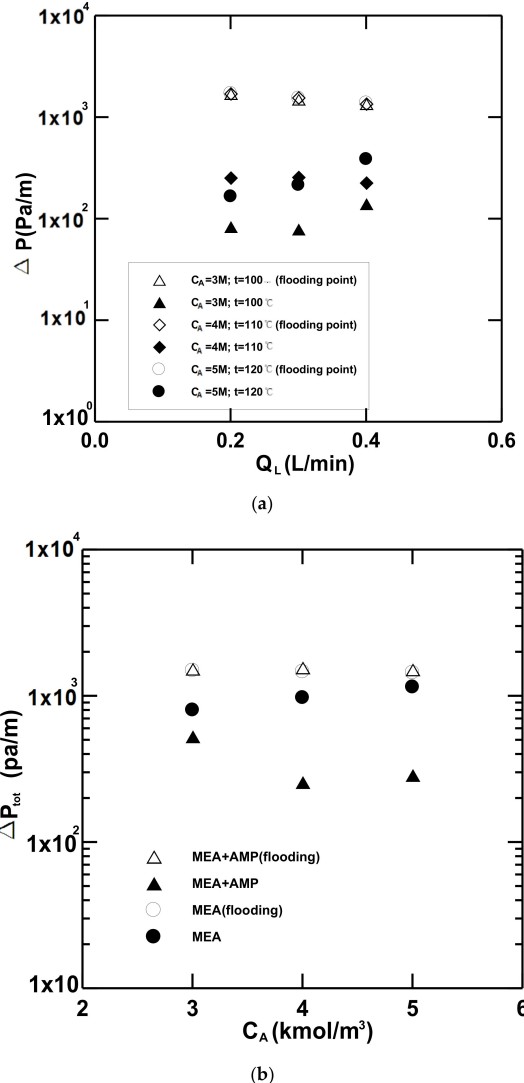

(a)

(b)

**Figure 4.** Effects of parameters on the pressure drops. (**a**) Effect of liquid flow rate on the pressure drop, (**b**) effect of concentration on the pressure drop.

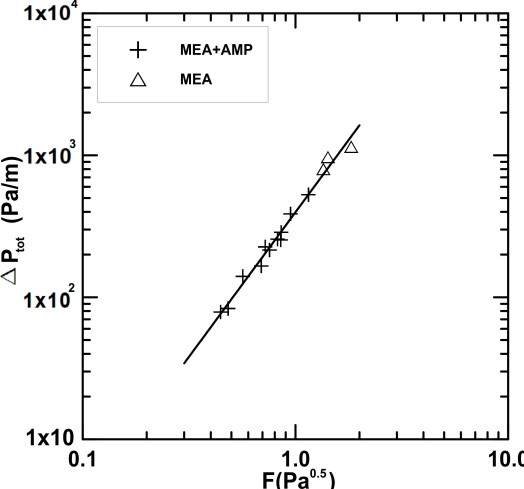

**Figure 5.** A plot of $\Delta P_{tot}$ versus F for both systems.

$R^2$ is 0.9845, indicating the reliability of the hydrodynamic model used here.

### 4.3. Evaluation Using Mass-Transfer Model

This study makes reference to VOC desorption within a packed bed, as proposed in literature [7], which is compared here. As reported by Ortiz-Del Castillo et al. [7], the specific surface area ranges from 141 to 492 $m^2/m^3$; the void fraction ranges from 0.90 to 0.98; the diameters range from $9 \times 10^{-4} - 1.7 \times 10^{-3}$ m; and the bed height and diameter are 2.8 m and 0.245 m, respectively. The L/D ratio is 11.4 when the θ-ring is applied in this study. Its specific surface area is 550 $m^2/m^3$; the void fraction is 0.97; the diameter is $3.273 \times 10^{-4}$ m; the bed height and diameter herein are 0.8 m and 0.05 m, respectively; and the L/D ratio is 16. As both L/D ratios are above 10 and the specific surface area and void ratio are close, this study applied this model for evaluation, including the mass transfer coefficient, HTU, and other data.

#### 4.3.1. Mass Transfer Data

First, the effective speeds, $u_{Ge}$ and $u_{Le}$, were calculated, as shown in Table S3. Individual liquid-phase and gas-phase mass transfer coefficients could be obtained from Equations (16)–(19). Then, equilibrium ratio m, the S value, the overall mass transfer coefficient $K_L a$, HTU, number of transfer unit (NTU), and other data could also be obtained [20], and such data are shown in Tables S3 and S4. The results show that $u_{Ge}$, $u_{Le}$, $k_G$, $k_L$, $a_e/a_p$, and $k_L a$, as listed in Table S2, are in the range of 0.5461–1.2264 m/s, 0.03388–0.04493 m/s, $6.793 \times 10^{-3}$–$1.146 \times 10^{-2}$ m/s, $2.052 \times 10^{-4}$–$2.553 \times 10^{-4}$ m/s, 0.162–0.220, and 0.0187–0.0272 1/s, respectively. While values m, H, G, L, S, and HTU, as presented in Table S4, are in the range of 1525–2752, 6211–9371 pa·$m^3$/kmol, 1.826–4.799 mol/min, 7.12–17.45 mol/min, 381.6–1053, and 0.0823–0.1329 m, respectively.

#### 4.3.2. Effect of Parameter on the Overall Mass-Transfer Coefficient

Figure 6a shows the plot of $k_L a$ versus $Q_L$ under different operating conditions. While it was found that $k_L a$ increased with the increase in $Q_L$, the effect of $C_A$ and T on $k_L a$ cannot be separated in this figure, which requires further discussion. Alternatively, the linear regression of $k_L a$ with $Q_L$, $C_A$, and $T_{reb}$ was required. A total of twelve sets of data (No.1–No.12) were used, as shown in the following equation:

$$k_L a = 3.45 \times 10^{-3} \exp(-\frac{210.97}{T_{reb}}) Q_L^{0.4693} C_A^{-0.1096} \tag{27}$$

The root mean relative error was 1.447%. The equation shows that $k_L a$ increased with the increase in temperature and $Q_L$, while $k_L a$ decreased with the increase in $C_A$. The parameter's importance sequence shows that $Q_L(1.384) > T_{reb}(1.108) > C_A(0.945)$. In order to compare this finding with the data estimated from model $(k_L a)_{mea}$ and evaluated according to empirical equation $(k_L a)_{cal}$, the plot of $(k_L a)_{cal}$ versus $(k_L a)_{mea}$ is shown in Figure 6b. It was found that most data were within a ±5% margin of error for $k_L a$, which demonstrates that the mass transfer model applied in this study is suitable to describe the mass transfer in a packed-bed stripper. In addition, this study found that $D_L$ is proportional to $k_L$, as shown in Equation (17), while $D_L$ is inversely proportional to $C_A$, as reported in the literature [10]. Therefore, $k_L$ is inversely proportional to $C_A$, and this result is reasonable, as shown in Equation (27). Figure 7 shows the effect of the solvent and concentration on $k_L a$; it will be discussed later. This was because the viscosity of MEA was 0.8–1.0 mpa-s, while the viscosity of the mixed amine was lower, 0.6–0.8 mpa-s. Therefore, the mass transfer coefficients were higher for mixed amines.

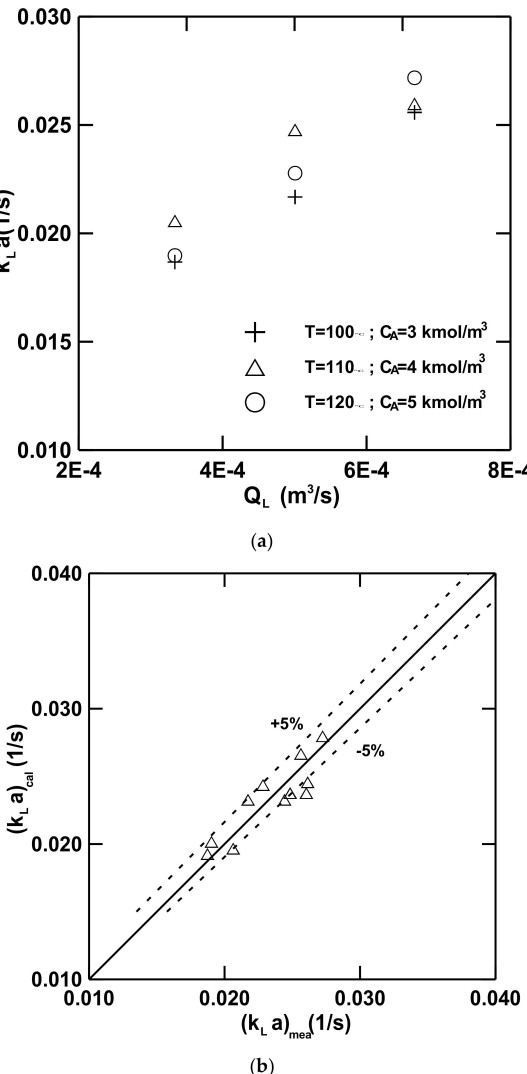

(a)

(b)

**Figure 6.** Effects of variables on mass transfer coefficient. (**a**) Effect of liquid flow rate on the kLa, (**b**) a plot of (kLa)emp versus (kLa)model.

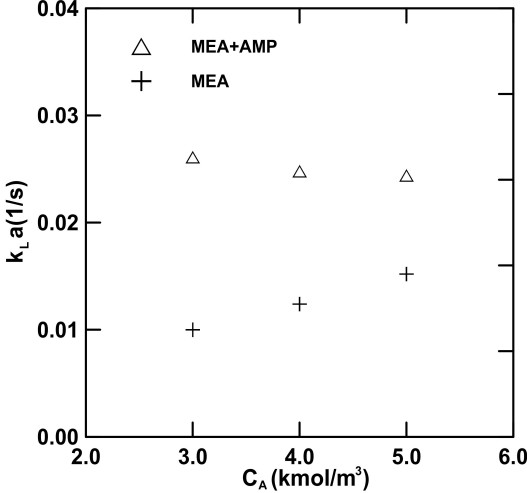

**Figure 7.** Effects of solvent and concentration on $k_L$a.

### 4.3.3. NTU and HTU

NTU and HTU can be estimated as reported in literature [7,9,20] or estimated as shown in in Equations (16)–(23). As $\frac{1}{k_L a_e} \gg \frac{1}{mk_G a}$, it indicates that the mass transfer is dominated by the liquid phase-side mass transfer, i.e., $K_L a_e \doteqdot k_L a_e$, which was in the range of 0.0187–0.0272 1/s. Furthermore, as HTU (m) ranges from 0.0823 to 0.1329 m, it indicates that the number of plates per meter range from 6 to 10; which is quite close to the number of plates per meter ranging from 7 to 10, as reported in the technical data provided by the Dixon ring manufacturer. This indicates that the mass transfer model applied herein is applicable to this study.

### 4.4. Comparison with Base-Line

This study used MEA as a base-line for comparison with the mixed solvent. Table 2 shows the data extracted from Tables S1–S4, including the $CO_2$ stripping rate ($\dot{m}_{CO_2}$), steam flow rate ($\dot{m}_s$), pressure drop, mass-transfer coefficient, and HTU. Under the same conditions for (Nos.10–12 and Nos.13–15), with the exception of $\dot{m}_{CO_2}$, the other items are all significantly different; $\dot{m}_s$ and $\Delta P_{tot}$ for MEA were much higher than the mixed solvent, which shows that AMP affected the solvent evaporation rate, and the pressure dropped. The range between $\Delta P_{tot}$ and $\Delta P_{tot,flood}$ shows that the operating range for a mixed solvent is higher than that for an MEA solvent. In addition, the $k_L a$ for a mixed solvent was twice as high as MEA, indicating that the size of the bed required for a mixed solvent is smaller than the MEA. It was also found that the HTU for MEA was two times higher than the mixed solvent, thus showing lower efficiency for the MEA solvent. From hydrodynamic and mass-transfer viewpoints, a mixed solvent is better than a MEA solvent.

### 5. Conclusions

This study successfully adopted both the hydrodynamic and mass-transfer models in a packed-bed stripper for estimation. Using transport balances in conjunction with the thermodynamic data, the outcome data can be determined at a steady-state condition. The effects of the process variables on pressure drop and mass-transfer coefficient are as follows. The total pressure drop increased with temperature and liquid flow rate, while the pressure drop decreased with solvent concentration. The dimensionless pressure drop at the flooding point was in the range of 0.142–0.181, which is in good agreement with practical experience. The flooding point can be effectively estimated using total pressure drop and the F-factor correlation equation. In addition, the effect of the process variables on the mass-transfer coefficient can be expressed in the correlation equation, which infers that the parameter's significant sequence is $Q_L > T_{reb} > C_A$, indicating that the liquid flow rate dominates in the mass-transfer. In addition, the number of plates per meter ranging from 6 to 10 was close to the reported values ranging from 7 to 10, as provided by the Dixon ring manufacturer. All the evidence indicated that both models applied herein are applicable to the desorption process of $CO_2$ gas in a packed-bed stripper. Finally, in comparison with the hydrodynamic and mass-transfer for both solvents, this study found that a mixed solvent is better than an MEA baseline solvent.

**Supplementary Materials:** The following are available online at https://www.mdpi.com/2227-9717/9/1/46/s1, Figure S1: Variation of temperature versus time, Figure S2: Temperature distribution in the Packed-bed, Figure S3: Lean loading changes with time.title, Table S1: Operating conditions and measured data obtained in this work, Table S2: Hydrodynamic data obtained in this work, Table S3: Mass transfer data obtained in this work, Table S4: Henry's law constant and HTU obtained in this work.

**Author Contributions:** Conceptualization, P.C.C. and M.-W.Y.; methodology, P.C.C.; data curation, P.C.C., M.-W.Y., Y.-L.L.; resources, P.C.C., M.-W.Y.; writing—Original draft preparation, P.C.C., Y.-L.L.; writing—Review and editing, P.C.C. and M.-W.Y.; funding acquisition, M.-W.Y. All authors have read and agreed to the published version of the manuscript.

**Funding:** This work was supported by the Taiwan Power Company, Taiwan.

**Institutional Review Board Statement:** Not applicable.

**Informed Consent Statement:** Not applicable.

**Acknowledgments:** The authors acknowledge the financial supports of the MOST in Taiwan (MOST-109-2221-E-262-004-) and Research Institute of Taiwan Power Company.

**Conflicts of Interest:** The authors declare no conflict of interest. The founding sponsors had no role in the design of the study; collection, analyses, or interpretation of the data; writing of the manuscript; or the decision to publish the results.

## Nomenclature

| | |
|---|---|
| $a_L$ | specific surface area of liquid ($m^2 \, m^{-3}$) |
| $a_e$ | effective specific surface area ($m^2 \, m^{-3}$) |
| $a_p$ | specific surface area of packings ($m^2 \, m^{-3}$) |
| C | total concentration ($kmol \, m^{-3}$) |
| $\overline{C}_{PA}$ | heat capacity of mixed amine ($kJ \, kg^{-1} \, K^{-1}$) |
| $C_A$ | concentration of amine ($kmol \, m^{-3}$) |
| $d_p$ | diameter of packings (m) |
| $d_L$ | diameter of liquid (m) |
| $D_G$ | diffusivity of gas ($m^2 \, s^{-1}$) |
| $D_L$ | diffusivity of liquid ($m^2 \, s^{-1}$) |
| F | F-factor defined in Equation (24) ($pa^{0.5}$) |
| G | gas molar flow rate ($kmol \, s^{-1}$) |
| $h_{dyn0}$ | dynamic hold up below the loading point (-) |
| $h_{dyn}$ | dynamic hold up (-) |
| $H_{CO_2}$ | Henry's law constant ($kpa \cdot m^3 \, kmol^{-1}$) |
| $k_L$ | liquid side mass-transfer coefficient ($ms^{-1}$) |
| $k_G$ | gas side mass-transfer coefficient ($ms^{-1}$) |
| $K_L a$ | overall mass-transfer coefficient ($ms^{-1}$) |
| L | liquid molar flow rate ($kmol \, s^{-1}$) |
| m | equilibrium ratio (mole fraction/mole fraction) |
| $\dot{m}_{CO_2}$ | stripping rate ($kgs^{-1}$) |
| $\dot{m}_s$ | steam flow rate ($kgs^{-1}$) |
| P | total pressure (pa) |
| $\Delta P_{dry}$ | specific dry pressure drop ($pa \, m^{-1}$) |
| $\Delta P_{tot}$ | specific pressure drop ($pa \, m^{-1}$) |
| $\Delta P_{tot,flood}$ | specific dry pressure drop at flooding ($pa \, m^{-1}$) |
| $Q_L$ | volumetric flow rate of liquid ($m^3 \, s^{-1}$) |
| $Re_G$ | Reynolds number for gas (-) |
| S | stripping factor (-) |
| $T_{reb}$ | temperature in the rebolier (K) |
| $T'_1$ | temperature at the column top (K) |
| $u_G$ | gas linear flow rate ($ms^{-1}$) |
| $u_L$ | liquid linear flow rate ($ms^{-1}$) |
| $u_{Ge}$ | effective gas velocity ($ms^{-1}$) |
| $u_{Le}$ | effective liquid velocity ($ms^{-1}$) |
| $x_{in}$ | mole fraction of liquid at inlet (-) |
| $x_{out}$ | mole fraction of liquid at outlet (-) |
| X | parameter in Equation (14)(-) |
| *Greek symbols* | |
| $\alpha_0$ | rich loading ($mol\text{-}CO_2 \, mol\text{-}amine^{-1}$) |
| $\alpha$ | lean loading ($mol\text{-}CO_2 \, mol\text{-}amine^{-1}$) |
| $\gamma$ | contact angel between the liquid and solid(deg) |
| $\varepsilon$ | void fraction (-) |
| $\mu_G$ | viscosity of gas ($mpa \cdot s$) |
| $\mu_L$ | viscosity of liquid ($mpa \cdot s$) |
| $\rho_G$ | density of gas phase ($kgm^{-3}$) |
| $\rho_L$ | density of liquid phase ($kgm^{-3}$) |
| $\sigma$ | surface tension ($Nm^{-1}$) |

## Abbreviations

| | |
|---|---|
| APM 2 | amino-2-methyl-1-propanol |
| HETP | height equivalent to a theoretical plate |
| HTU | height of transfer unit |
| MEA | monoethanolamine |
| NTU | number of transfer unit |

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
