# Peer review of "Hydrodynamic and Mass Transfer in the Desorption Process of CO2 Gas in a Packed-Bed Stripper"

_processes, doi:10.3390/pr9010046_

Round 1

Reviewer 1 Report

Dear Authors,

It has been very interesting for me to review this study in which hydrodynamic and mass transfer data applied to CO2 desorption in a mixed solvent loaded packed-bed stripper with  mixed solvent (MEA+AMP) have been determined. Adapting the models included in the literature to the case study and exploring the different parameters involved.

The authors provide powerful results on the effects of process variables on pressure drop and mass transfer coefficient areas. In this study concludes that the significant sequence of the main parameters is the volumetric liquid flow rate, the temperature in the rebolier and the concentration of amines studied, and indicating that the liquid flow rate dominates in mass-transfer. Finally, this study shows that a mixed solvent is better than an MEA baseline solvent.

Therefore, the topic of this paper is appropriate for publication in the journal Processes. However, authors must do a full review of the manuscript and adapt it to the guidelines of the journal. Typographical and style corrections such as:

Line 103. Table header and table interrupted by a line of text. Table 1 must be improved. Adapt the content to a table.

Lines 105-106. Equations 16 to 22 or Equations 16 to 23?

Table 3. Adapt it to the guidelines of the journal.

Lines 155 and 156: T’1 or T1’ as shown Figure 3?

Lines 188 and 190: missing spaces.

Figure 4. Set the font size of the figure's subtitles. In addition, the figure below should be (b) instead of (a). The two figures are shown as (a).

Lines 203-213: Set as guidelines rules.

Line 210: Figure 4 (b) or Figure 5.

Lines 220 and 222. Specific surface area units m2/m3? Please, international units.

In particular, I consider it important for a successful review to have the supplementary material available. I do not find Tables S1 to S4 and Figures S1 to S3. I guess that some of these tables and figures should be included in the manuscript, but this could be important.

Regards. 

Author Response

It has been improved. Please see the new version in red words. Thank you 

Reviewer 2 Report

The paper deals with the hydrodynamic and mass transfer in the desorption of CO2 in packed bed strippers. The paper is very actual and well written. The reviewer opinion is very positive aboiut both methods and the obtained results, thus he suggests to accept it in the present form.

A minor remark: some references style should be fixed.

Author Response

Dear Reviewer:

  Your comments are very useful to me. Thank you 
